

# Taxifolin protects rat against myocardial ischemia/reperfusion injury by modulating the mitochondrial apoptosis pathway

Zhenqiu Tang[1,*], Chunjuan Yang[2,3,*], Baoyan Zuo[4], Yanan Zhang[1], Gaosong Wu[1], Yudi Wang[1] and Zhibin Wang[1]

[1] Key Laboratory of Chinese Materia Medica, Heilongjiang University of Traditional Chinese Medicine, Harbin, Heilongjiang, China
[2] College of Pharmacy, Harbin Medical University, Harbin, Heilongjiang, China
[3] Beijing Shunyue Technology Co., Ltd., Beijing, China
[4] School of Pharmacy, Shenyang Pharmaceutical University, Shenyang, Liaoning, China
* These authors contributed equally to this work.

## ABSTRACT

**Background:** Taxifolin (TAX), is an active flavonoid, that plays an underlying protective role on the cardiovascular system. This study aimed to evaluate its effect and potential mechanisms on myocardial ischemia/reperfusion (I/R) injury.

**Methods:** Healthy rat heart was subjected to I/R using the Langendorff apparatus. Hemodynamic parameters, including heart rate, left ventricular developed pressure (LVDP), maximum/minimum rate of the left ventricular pressure rise ($+\mathrm{d}p/\mathrm{d}t_{max}$ and $-\mathrm{d}p/\mathrm{d}t_{min}$) and rate pressure product (RPP) were recorded during the perfusion. Histopathological examination of left ventricular was measured by hematoxylin-eosin (H&E) staining. Creatine kinase-MB (CK-MB) and lactate dehydrogenase (LDH) activities in the effluent perfusion, and the levels of malondialdehyde (MDA), superoxide dismutase (SOD), and glutathione peroxidase (GSH-PX) in the tissue were assayed. Apoptosis related proteins, such as B-cell lymphoma-2 (Bcl-2), Bcl2-associated X (Bax), and cytochrome c (Cyt-c) were also assayed by ELISA. Western blot was employed to determine apoptosis-executive proteins, including caspase 3 and 9. Transferase-mediated dUTP-X nick end labeling assay was performed to evaluate the effect TAX on myocardial apoptosis.

**Results:** Taxifolin significantly improved the ventricular functional recovery, as evident by the increase in LVDP, $+\mathrm{d}p/\mathrm{d}t_{max}$, $-\mathrm{d}p/\mathrm{d}t_{min}$ and RPP, the levels of SOD, GSH-PX were also increased, but those of LDH, CK-MB, and MDA were decreased. Furthermore, TAX up-regulated the Bcl-2 protein level but down-regulated the levels of Bax, Cyt-c, caspase 3 and 9 protein, thereby inhibits the myocardial apoptosis.

**Discussion:** Taxifolin treatment remarkably improved the cardiac function, regulated oxidative stress and attenuated apoptosis. Hence, TAX has a cardioprotective effect against I/R injury by modulating mitochondrial apoptosis pathway.

Corresponding author
Zhibin Wang,
wzbmailbox@hljucm.net

## INTRODUCTION

Ischemic heart disease is a threat to human health. Extracorporeal circulation and coronary bypass surgery are usually employed to improve myocardial ischemia after myocardial infarction occurs. The timely restoration of blood flow to the ischemic myocardium (reperfusion) became the standard treatment for these patients. However, reperfusion may cause additional heart damage. This condition is referred to as cardiac ischemia/reperfusion (I/R) injury (*Braunwald, 2012*). The reperfusion of ischemic tissues is often associated with microvascular dysfunction. The mechanisms may involve the release of oxygen radicals and inflammatory mediators (*Carden & Granger, 2000*). Nowadays, tissue injury induced by I/R is a major factor, which often cause death. During myocardial I/R injury, cardiomyocytes undergo death at an increased frequency, mainly including necrosis and apoptosis (*Gottlieb et al., 1994*). Apoptosis involves programmed cell death, which is the vital pathological process in acute reperfusion injury (*Konstantinidis, Whelan & Kitsis, 2012*). When the amount of cardiomyocyte decreases, the heart may undergo ventricular remodeling, compensatory cardiac hypertrophy, and eventually lead to heart failure (*Pangonyte et al., 2008*; *Du et al., 2010*). Therefore, exploring the detailed mechanisms that trigger cardiomyocyte death and the means to prevent it during I/R injury is still a public issue.

In I/R injury, morphological changes in myocardial tissue can be observed, including microvascular damage and myocardial cell edema. The symptoms of I/R include myocardial enzyme release, arrhythmias and weak systolic function (*Naito et al., 2000*). Generally, this reperfusion damage is caused by increased free radical activity. When circulating blood decreases, the level of oxygen supply cannot sustain the oxygen demand by cardiomyocytes, and the aerobic metabolism turns into anaerobic metabolism (*Giordano, 2005*). Anaerobic metabolism leads to the production of lactic acid which results in disturbances in ionic homeostasis. A timely reperfusion is crucial for the recovery of an ischemic myocardium, but by the sudden re-appearance of circulating blood to the dying myocardium, oxygen species (ROS) will be produced as a response to hyperoxia which can worsen the functional situation of the organization (*Akhlaghi & Bandy, 2009*).

Flavonoid is the most prevalent class of naturally occurring compound and is ubiquitous in woody and herb plants. It exerts multiple biochemical properties and wide pharmacological effects (*Moon, Wang & Morris, 2006*). Epidemiological studies have shown that flavonoid is associated with a reduced risk of cardiovascular diseases (*Raj Narayana et al., 2001*; *Bjorklund et al., 2017*). Fisetin, a plant-derived bioflavonoid, significantly attenuated I/R-induced tissue injury, blunted the oxidative stress, and restored mitochondrial structure and function (*Shanmugam et al., 2018*). Quercetin has been demonstrated to improve post ischemic recovery of the isolated heart of rats after global ischemia (*Barteková et al., 2010*). Taxifolin (TAX) exerts anti-inflammatory effects and prevents oxidative stress-induced injury in human endothelial cells (*Guo et al., 2015*) and rat peritoneal macrophages (*Arutyunyan et al., 2016*). It also possesses free radical scavenging, antioxidant and anti-inflammatory effects (*Sun et al., 2014*; *Xie et al.,*

*2017*). TAX is structurally similar to quercetin. Hence, we suspect that it also has a beneficial effect on the cardiovascular system. Recent studies demonstrated that TAX could inhibit cardiac hypertrophy and attenuate ventricular fibrosis after pressure overload. These beneficial effects were at least mediated by suppressing oxidative stress and the excess production of ROS (*Guo et al., 2015*; *Sun et al., 2014*). However, the potential of TAX for I/R protection remains unclear. TAX is a potential candidate for the prevention or treatment for I/R injury. However, the influence of TAX on the injury of I/R in isolated rat hearts has not been reported. In this study, we aimed to evaluate the cardioprotective effects of TAX and investigated the mechanisms underlying these effects in isolated hearts of rats.

Myocardial ischemic events are unpredictable. The clinical application of pre-conditioning drugs is limited. Therefore, researchers turned to a new endogenous protective strategy, which is post-conditioning. In 2006, the protective effect of post-conditioning in ischemic reperfusion on cerebral ischemia was first reported (*Zhao, Sapolsky & Steinberg, 2006*). Subsequent studies have validated the protective effect of post-conditioning in various global ischemia models ex vivo and ischemia and hypoxia models in vitro. Flavonoids reduce the injury of myocardial ischemia on perfusion in a post-conditioning way (*Wang et al., 2017*; *Xuan & Jian, 2016*). Therefore, we adopted the post-conditioning method in this study.

## MATERIALS AND METHODS

### Experimental animals and treatment

Male standard deviation (SD) rats (280–300 g each) were obtained from the Laboratory Animal Center of Heilongjiang Medicine University Medical (License Number: SCXK (hei) 2013-004). The rats were housed under standard conditions with natural light (12 h) and dark (12 h) at 22 ± 2 °C. Rats were fed with common laboratory chow and allowed to drink tap water ad libitum during the experimental period. The investigation conformed to Guide for the Care and Use of Laboratory Animals (revised, 1996, http://dels.nas.edu/resources/static-assets/ilar/miscellaneous/GUIDE1996.pdf). All animal experiments were approved by the College of Pharmacy of Heilongjiang University of Chinese Medicine, Animal Ethics Committee (Approval number: SYXK (hei)-2013-012).

### Reagents and antibodies

Taxifolin (purity ≥ 98%) was purchased from Sigma-Aldrich (St. Louis, MO, USA). Creatine kinase-MB (CK-MB), lactate dehydrogenase (LDH), malondialdehyde (MDA) glutathione peroxidase (GSH-PX), and superoxide dismutase (SOD) assay kit were obtained from Nanjing Jiancheng Bioengineering Institute (Jiangsu, China). The enzyme-linked immunosorbent assay kit about mitochondrial apoptosis (B-cell lymphoma-2 (Bcl-2), Bcl2-associated X (Bax), and cytochrome c (Cyt-c)) was obtained from Cloud-Clone Corp (Houston, TX, USA). Monoclonal primary antibodies anti-β-actin, anti-active caspase 3 and 9 were purchased from Abcam (Cambridge, MA, USA). Fluorescent-labeled goat anti-rabbit IgG secondary antibody was obtained from LI-COR Biosciences (Lincoln, NE, USA).

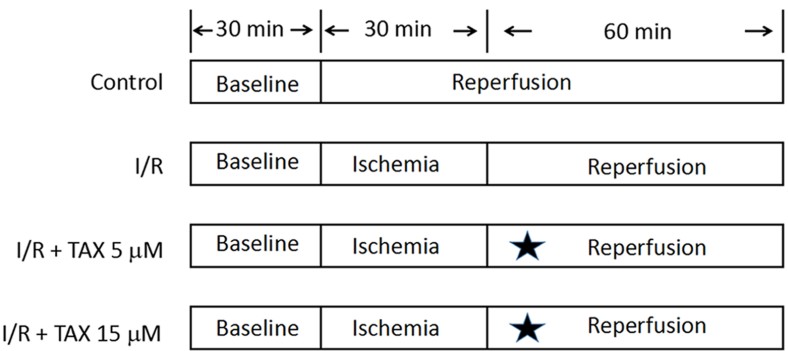

**Figure 1 Experimental protocol.** Schematic diagram of experimental protocol. Normal control group (Control); myocardial I/R control group (I/R); I/R + TAX treatment group (TAX 5 µM); I/R + TAX treatment group (TAX 15 µM).

## Experimental protocol

Taxifolin was dissolved in ethanol (15.21, 45.63 mg/mL) and then transferred into Krebs–Henseleit (K–H) solution. Ethanol solution (0.1 mL) was added into 1 L K–H solution. The final concentration was 5 or 15 µM. Rats were randomly divided into four groups ($n = 8$): Normal control group (Control); Myocardial I/R control group (I/R); I/R + TAX 5 µM treatment group (TAX 5 µM); I/R + TAX 15 µM treatment group (TAX 15 µM). The experimental protocol is shown in Fig. 1. Control group: The hearts were subjected a continuous perfusion of K–H solution for 120 min. I/R group: The hearts were perfused for 30 min to stabilization. Subsequently, global ischemia was performed at 37 °C for 30 min, followed by reperfusion with K–H solution for 60 min. TAX 5 µM group: The hearts were perfused for 30 min to stabilization. Subsequently, global ischemia was performed for 30 min at 37 °C, followed by reperfusion with five µM of TAX-saturated K–H solution for 60 min. TAX 15 µM group: The hearts were perfused for 30 min to stabilization. Subsequently, global ischemia was performed for 30 min, followed by reperfusion perfused with 15 µM of TAX-saturated K–H solution for 60 min.

## Langendorff preparation

After anesthetization via intraperitoneal injection (chloral hydrate solution, 300 mg/kg), rat hearts were quickly removed and subsequently perfused in the Langendorff apparatus. The perfusion was performed for 30 min in a modified K–H buffer gassed with 95% $O_2$ and 5% $CO_2$ at a constant flow velocity and constant temperature (37 °C). The composition of K–H buffer as the following (mM): NaCl 118, $MgSO_4$ 1.2, KCl 3.2, $NaHCO_3$ 25, $KH_2PO_4$ 1.18, $CaCl_2$ 2.5, and glucose 5.5. After equilibration, 30 min global ischemia was induced followed by 60 min of reperfusion. The control group utilized the same protocol, but no ischemic induction was used. Water-filled balloon that is inserted into the left ventricular cavity was used to monitor hemodynamic parameters. The left ventricular end-diastolic pressure was maintained at 5–10 mmHg by adjusting the size and position of the balloon. The whole procedure was completed within 2 min.

The inclusion criteria of experimental samples were a heart rate (HR) of >250 beats/min and a left ventricular developed pressure (LVDP) of >75 mmHg in equilibrium period. In the experiment, we prepared 12 rats in each group with an average of three or four failures. Finally, only eight rats from each group were used in the subsequent experiment. The hemodynamic parameters were recorded during perfusion, including HR, LVDP, the maximum/minimum rate of left ventricular pressure rise ($+dp/dt_{max}$ and $-dp/dt_{min}$), which are important indices to evaluate the left ventricular systolic and diastolic function. Rate pressure product (RPP) = HR $\times$ LVDP. After 60 min of reperfusion, heart tissue was taken from Langendorff apparatus. The left ventricles of three hearts were cut from each group and then fixed in 10% neutral formalin. The other tissues were immediately placed in the freezer at $-80\ °C$.

## Histopathological evaluation of left ventricle sections

For histopathological examination, the cut left ventricle of heart tissues were fixed in 10% neutral formalin at room temperature. After 2 h, the tissue piece was embedded in paraffin. Next, the piece was cut into three μm thick tissue sections, after which it was subjected to hematoxylin-eosin (H&E) staining. At least three samples from each group were evaluated. The tissue sections were visualized under light microscope (Olympus BX60, Tokyo, Japan).

## Estimation of cardiac damage

In present study, heart tissue injury was assessed by determining the concentration of LDH and CK-MB in the perfusate. The LDH and CK-MB content in the perfusate were measured using the assay kit following the manufacturer's instructions. Samples of the perfusate were collected from the isolated heart at 25, 63, 90, and 120 min of perfusion.

## Measurements of anti-oxidant indices

The hearts tissue was cut into small pieces of tissue and then was ground with lysate buffer by using a glass homogenizer. Supernatant of tissue homogenate was frozen for each tissue analysis. MDA, SOD, and GSH-PX activity were assessed using commercial ELISA kits following the manufacturer's instructions. All enzyme activities were normalized to the total protein concentrations, which were determined using a bicinchoninic acid protein assay kit (Beyotime, Shanghai, China).

## Estimation of Cyt-c, Bcl-2, and Bax levels

Following the instructions of nuclear and cytoplasmic protein extraction kit (Beyotime, Shanghai, China), heart tissue samples were weighed, minced into small pieces and homogenized using a glass homogenizer on ice (w:v = 1:30, one mL lysis buffer was added in 30 mg tissue sample). The homogenates were centrifuged at $1,500 \times g$ for 5 min at $4\ °C$ and the cytoplasmic protein was obtained from the supernatant. The Cyt-c, Bcl-2, and Bax protein levels were measured according to the manufacturer's instructions of commercial kits through an ELISA-type method (Cloud-Clone Corp, Houston, TX, USA).

## Western blotting analysis

Myocardial tissue samples were lysed with RIPA buffer containing protease inhibitors for 15 min on ice. The total lysates were clarified by centrifugation, and supernatants were collected. Protein samples (20–25 mg per lane) were loaded on the gels and then separated by 10% sodium dodecyl sulfate-polyacrylamide gel electrophoresis under reducing conditions and transferred onto the nitrocellulose membrane (Roche, Mannheim, Germany). The membrane was washed with PBS with 0.1% Tween-20 (PBST) and blocked with 5% skim milk in shaking table for 2 h at room temperature. Then the membrane was washed with PBST and incubated with antigen-specific rabbit IgG antibodies (anti-caspase 3 and 9, anti-β-actin; Abcam) diluted to 1:1,000 in PBST. Next, the membrane was washed with PBST and incubated with fluorescent-labeled goat anti-rabbit secondary antibodies (LI-COR, Lincoln, NE, USA) diluted to 1:2,500 in PBST for 2 h at 4 °C. The target protein bands were scanned using the blot imaging system GelLogic 212 PRO (Carestream, Rochester, NY, USA) after washing with PBST. The obtained images were quantified as the final results by image J 1.4.3 (www.imagej.nih.gov/ij). The results were expressed as the fold induction, which were than compared with the normal control.

## TUNEL assay

Heart sections of three μm thickness were obtained using a microtome. The sections were deparaffinized in xylene and rehydrated in concentration gradient of ethanol (100%, 95%, 75%). Following this, sections were then incubated with proteinase K and with 30% $H_2O_2$ to enhance tissue permeability and diminish any endogenous peroxidase activity respectively. Apoptosis was determined using a terminal deoxynucleotidyl transferase-mediated dUTP-X nick end labeling (TUNEL) assay kit (Roche, Basel, Switzerland) according to the manufacturer's protocol. The sections were incubated with complete labeling reaction buffer for 60 min and antibody solution for 30 min. Chromogenic reaction was visualized using 3,3-diaminobenzidine. Sections were visualized under light microscope (Olympus BX60, Tokyo, Japan).

## Statistical analysis

SPSS16.0 for Windows (SPSS Inc., Chicago, IL, USA) was used for statistical analysis. All data were expressed as mean ± SD. For comparisons between groups, the one-way ANOVA or student $t$-test was used where appropriate. Statistical differences were considered significant at $P < 0.05$. [#]$P < 0.05$, and [##]$P < 0.01$ vs. the control group. [*]$P < 0.01$ and [**]$P < 0.001$ vs. the I/R group.

# RESULTS

## Effects of TAX on cardiac parameters of isolated hearts

We examined whether TAX could protect the hearts of rat against ex vivo I/R injury. Results showed no obvious alteration in the average HR of isolated heart during reperfusion with or without TAX. In addition, no significant HR change was observed between I/R and normal groups during the 30 min of ischemia and 60 min of reperfusion (Fig. 2A). After

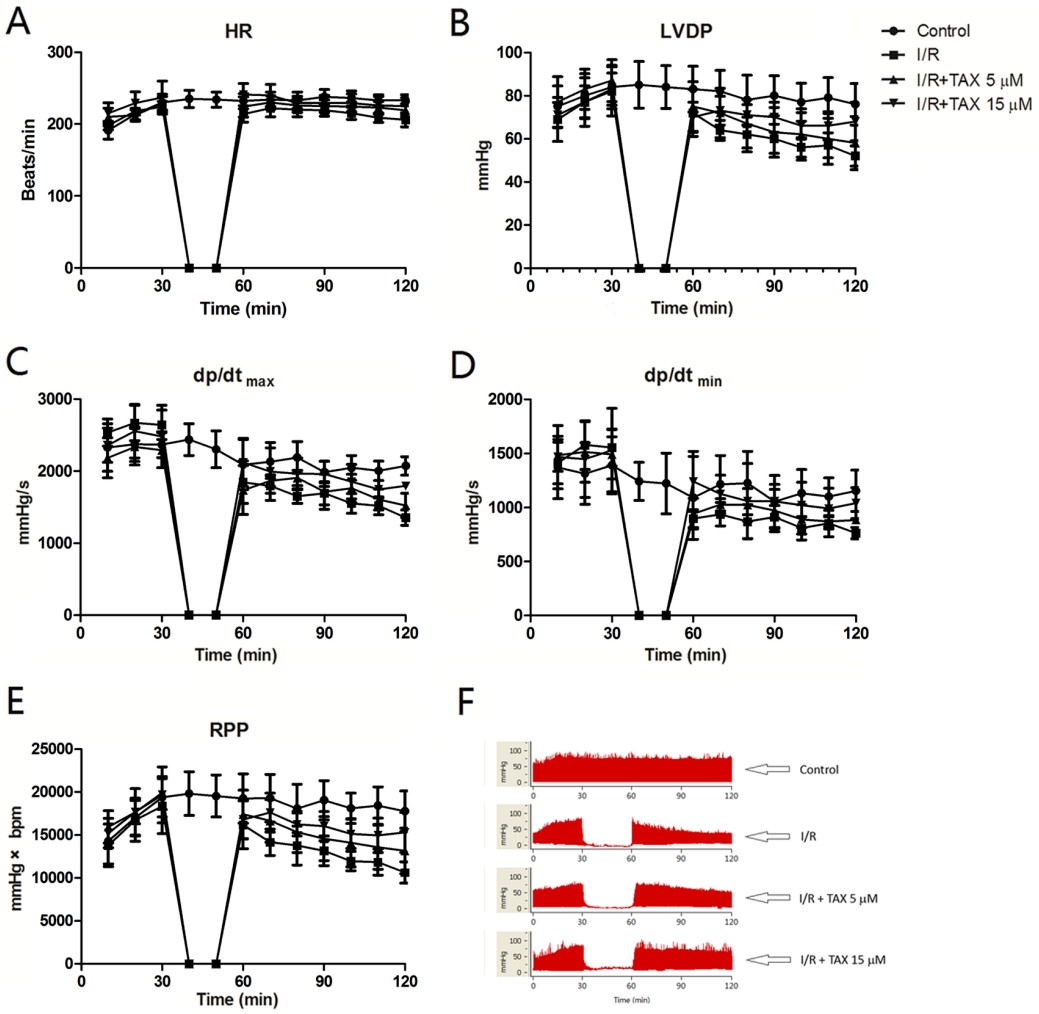

**Figure 2 TAX treatment improved the cardiac function recovery of rats during myocardial I/R injury in vitro model.** (A) Heart rate (HR, beat/per min); (B) left ventricular developed pressure (LVDP, mmHg); (C) maximum rate of left ventricular pressure (+$dp/dt_{max}$, mmHg/s); (D) minimum rate of increase of left ventricular pressure (−$dp/dt_{min}$, mmHg/s); (E) Rate pressure product (RPP, mmHg × bpm); (F) representative left ventricular pressure records in experimental protocol form different experiment groups.

reperfusion, LVDP, +$dp/dt_{max}$, and −$dp/dt_{min}$ from different treatment groups decreased in varying degrees. For instance, LVDP was significantly increased in the TAX 15 μM group compared with the I/R group at the end periods of reperfusion (LVDP = 68 vs. 52 mmHg, $P < 0.05$, Fig. 2B). In comparison with the I/R group, treatment with 15 μM TAX significantly improved the RPP in rat at 60 min of reperfusion (RPP = 15,294 mmHg × beats/min in TAX vs. 10,643 mmHg × beats/min in I/R, $P < 0.01$, Fig. 2E). Results showed that TAX treatment improved the cardiac function recovery of rats during myocardial I/R injury.

## TAX down-regulated the release of LDH and CK-MB

At different time points of perfusion, heart effluents were collected. The LDH level in the whole perfusion process was not conspicuously altered in the control group. The

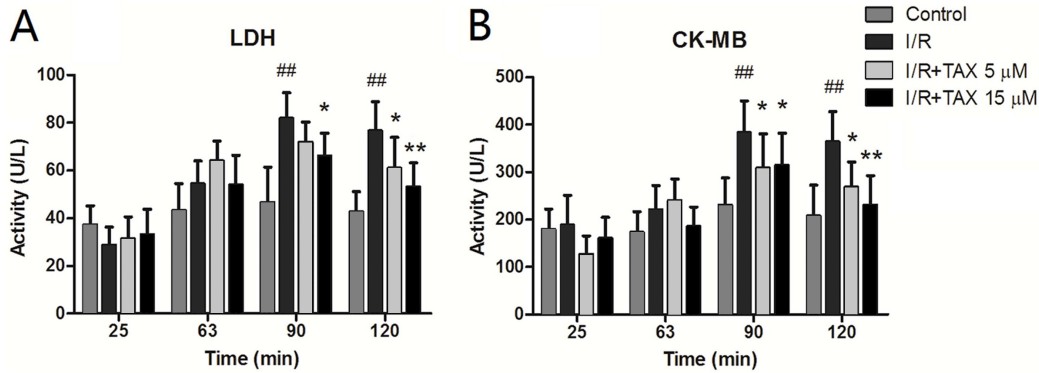

**Figure 3 Effect of TAX on injury of cardiomyocytes by measurement of LDH and CK-MB.** (A) The activity of perfusate LDH; (B) the activity of perfusate CK-MB. The levels of perfusate LDH and CK at different time points in the control, I/R and TAX-treat groups (5 and 15 $\mu$M) are shown. $^{\#\#}P < 0.01$ compared with the control group; $^{*}P < 0.05$ and $^{**}P < 0.01$ compared with the I/R group; U/L: international enzyme activity unit per liter.     

perfusate LDH activity of the I/R group was improved compared with that of the control group after reperfusion and was significantly increased at 60 min of reperfusion ($P < 0.01$). TAX highly reduced the LDH levels compared with the I/R group at 60 min of reperfusion (Fig. 3A). CK-MB release was similar to the LDH release. The expression in the whole perfusate process was not conspicuously changed in the control group. In the perfusion stabilization, no significant difference was observed in the CK-MB level among the four groups. However, the levels in the I/R group were markedly higher after 30 min of reperfusion compared with perfusion stabilization ($P < 0.05$). Interestingly, the CK-MB release in both the TAX 5 and 15 $\mu$M groups was significantly decreased at the end of reperfusion compared with the I/R group ($P < 0.05$ or $P < 0.01$) (Fig. 3B). These results indicated that TAX could protect the cardiac function against I/R injury.

## Effect of TAX on myocardial morphology

Histopathological examination of myocardial tissue was assessed by H&E staining. Typical micrographs of the myocardial structure are shown in Fig. 4. In the control group (Fig. 4A), the morphology of the myocardial tissue was normal. Cardiomyocytes are arranged closely, the intercellular space is small, and edema does not exist between cells. By contrast, the I/R group (Fig. 4B) showed degenerated muscle fibers and obvious contraction band, severe obvious cells edema, many infiltrated inflammatory cells. Figure 4C shows that the TAX 5 $\mu$M group maintained the myocardium with only slightly irregularly arranged fibers and a few contraction bands. Figure 4D shows that the TAX 15 $\mu$M group showed orderly cardiomyocytes but a few cell dissolution and degeneration. Results showed that treatment with 15 $\mu$M TAX significantly reduced I/R injury compared with I/R group.

## Effect of TAX on I/R-induced oxidative stress in the myocardium

To explore the cardio-protective mechanism of TAX, the effects of TAX on SOD, GSH-PX, and MDA activity were investigated in myocardial tissue in response to I/R injury.

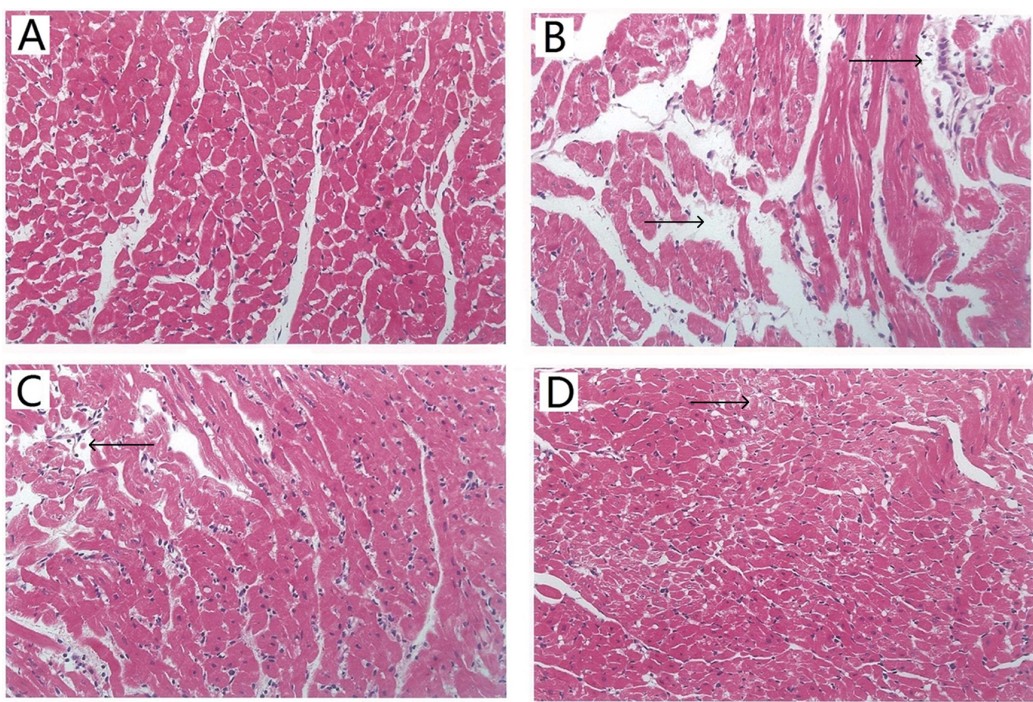

**Figure 4 Representative micrographs of HE staining results in various experimental groups.**
(A) Control group; (B) I/R group; (C) TAX 5 μM group; (D) TAX 15 μM group, $n = 3$ per group,
(magnification, ×400), (←) and (→) tissue damage and edema.

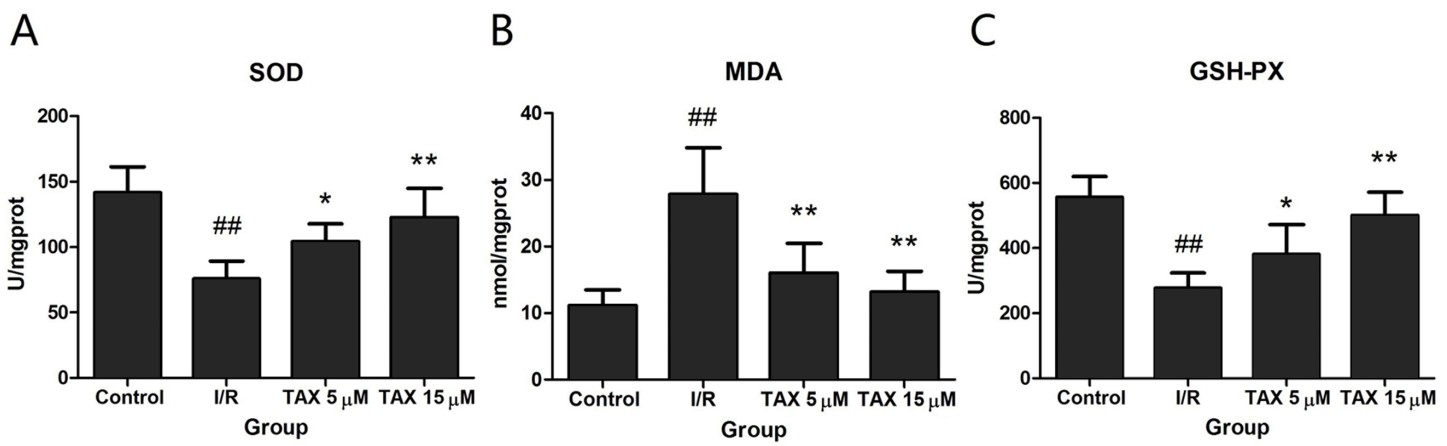

**Figure 5 Effect of TAX on cardiac the activity of SOD and GSH-PX, contents of MDA.** (A) The activity of SOD; (B) the content of MDA; (C) the
activity of GSH-PX. Values are presented as mean ± SD. $^{\#\#}P < 0.01$ compared with the control group; $^{*}P < 0.05$ and $^{**}P < 0.01$ compared with the I/R
group. U/mgprot: international enzyme activity unit per milligram tissue protein.

Figure 5 shows that TAX 15 μM group, the SOD and GSH-PX activity were increased
significantly compared with those in the I/R group ($P < 0.01$), whereas no significant
difference was observed in the TAX 5 μM group. Conversely, these TAX treatment
groups showed that MDA production was significantly reduced ($P < 0.01$) compared with
the I/R group.
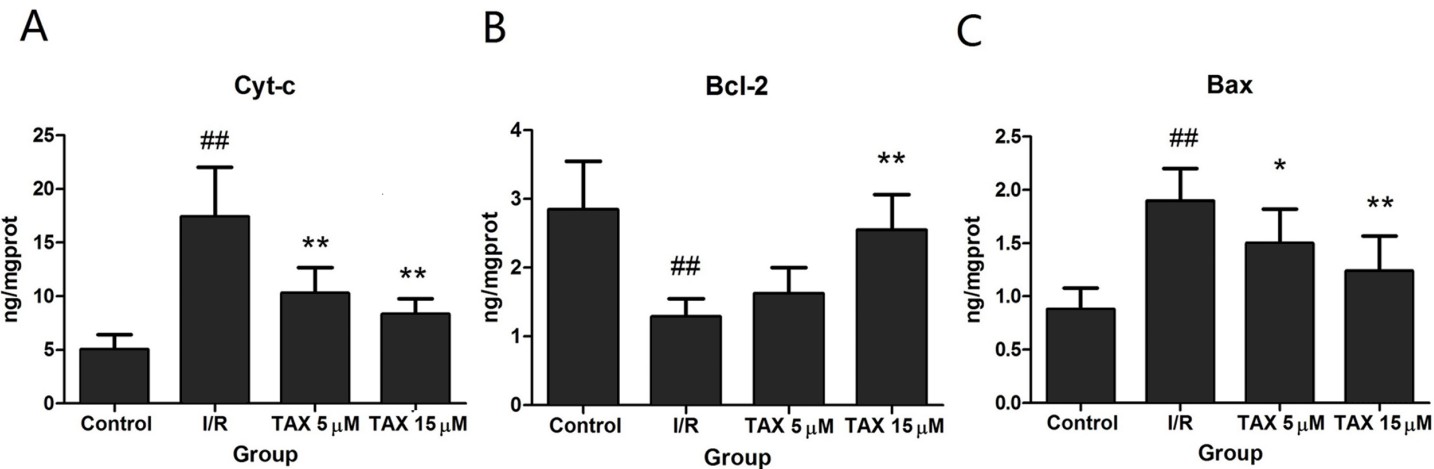

**Figure 6 Effect of TAX on the expression of Cyt-c, Bax, and Bcl-2 protein.** (A) The expression of Cyt-c protein; (B) the expression of Bcl-2 protein; (C) the expression of Bax protein. $^{##}P < 0.01$ vs. the control group; $^{*}P < 0.05$, $^{**}P < 0.01$ vs. the IR group. ng/mgprot indicate the nanogram level of the target protein per milligram total protein.

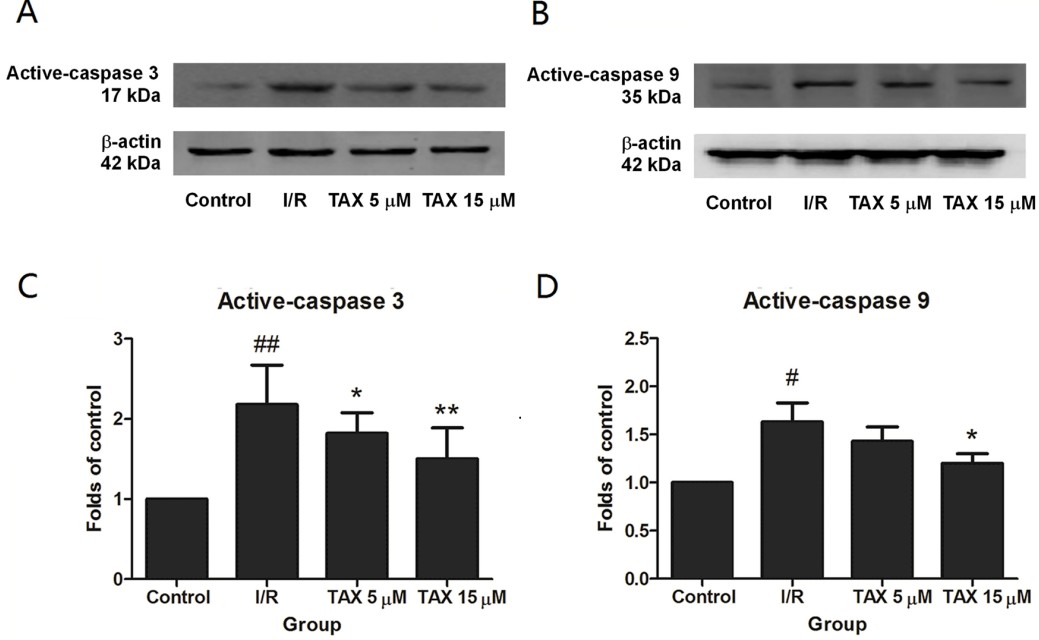

**Figure 7 The changes in the levels of caspase 3, and caspase 9 at the end of reperfusion.** (A) Western blot image of caspase 3; (B) western blot image of caspase 9; (C) the relative expression of caspase 3 protein; (D) the relative expression of caspase 9 protein. $^{#}P < 0.05$ and $^{##}P < 0.01$ compared with the control group, $^{*}P < 0.05$ and $^{**}P < 0.01$ compared with the I/R group.

## TAX protects myocardial cell from I/R-induced mitochondrial damage

To evaluate if the effect of TAX is mediated through attenuation of the mitochondrial damage, we determined Cyt-c in cytosol. Figure 6A shows that I/R increased the cytosolic Cyt-c levels ($P < 0.01$). By comparison, TAX at different doses could reduce I/R-induced increase of Cyt-c levels ($P < 0.01$). The result suggested that TAX attenuated

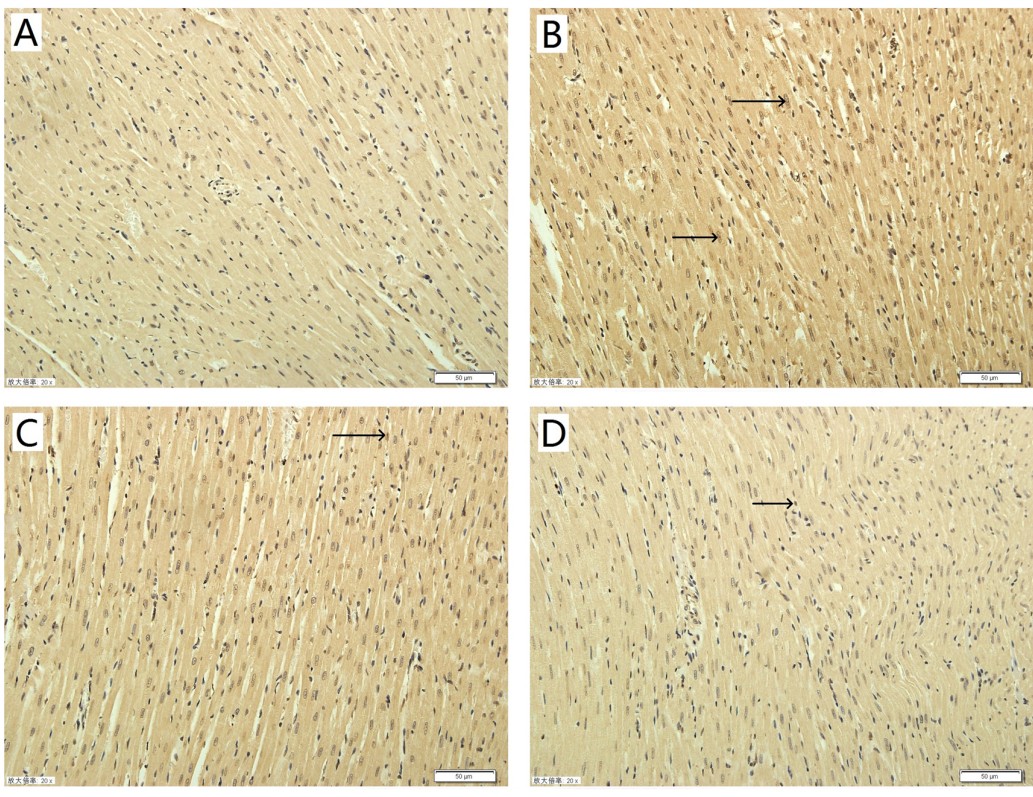

**Figure 8 Effect of taxifolin on cardiomyocytes apoptosis in different experimental groups.**
(A) Control group; (B) I/R group; (C) I/R + TAX 5 μM group; (D) I/R + TAX 15 μM group. $n = 3$ per group, 20×, scale bar 50 μm, (→) positive stain.

I/R induced Cyt-c release to the cytosol. The Bcl-2 family of proteins modulates the release of Cyt-c by regulating mitochondrial transmembrane potential. We also determined Bcl-2 and Bax protein levels. Compared with the control group, I/R down-regulated Bcl-2 but up-regulated Bax protein expression (Figs. 6B and 6C). TAX increased Bcl-2 levels but decreased Bax levels compared with I/R. These results indicate that TAX may protect mitochondria against apoptosis by regulating the expression of the Bcl-2 family proteins.

## TAX attenuates myocardial I/R-induced apoptosis

To explore the potential mechanism of TAX in rats subjected to I/R-induced myocardial injury, we investigated the protein expressions of active-caspase 3 and 9. Compared with the control group, the expression of active-caspase 3 was up-regulated in the I/R group. While compared with the I/R group, the TAX treatment group significantly reduced the level of active-caspase 3 (Fig. 7C). In the TAX treatment group, the expression of active-caspase 9 was down-regulated in 15 μM ($P < 0.05$), but did not change significantly in five μM (Fig. 7D). In addition, TUNEL assay was performed to evaluate the effect of TAX on myocardial apoptosis. An increased number of TUNEL positive cells were observed in I/R group in comparison to control group. Furthermore, a lesser number of TUNEL positive cells were present in the TAX treatment group (Fig. 8). Results indicated that TAX inhibited apoptotic level in heart injury induced by I/R.

## DISCUSSION

Growing evidence indicates a therapeutic action of TAX in cardiovascular disease. However, the implications of TAX in I/R injury remain unclear. To the best of our knowledge, this study was the first to evaluate the cardioprotective effects of TAX in isolated rat heart subjected to I/R injury. We demonstrated that an important role of TAX in improving cardiac function, and inhibiting oxidative stress and apoptosis in a model of I/R injury in vitro.

In the late 19th century, Oscar Langendorff pioneered the isolated perfused heart model. Since then, the procedure has been used to probe the pathophysiology of I/R with the dawn of molecular biology (*Bell, Mocanu & Yellon, 2011*). Today, the Langendorff heart assay is a predominant technique in vitro, which is used in physiological and pharmacological research. It allows the examination of cardiac contractile strength and HR without complications of an intact animal (*Herr, Aune & Menick, 2015*). Therefore, we determined the effect of TAX to the cardiac function parameters of isolated heart in myocardial I/R injury by using the Langendorff equipment. Cardiac functions mainly depend on the contraction and relaxation properties of the ventricular muscle. Changes in cardiac function are strongly linked to the severity of I/R injury (*Mehdizadeh et al., 2013*). The data from our analyses showed that I/R can cause marked myocardial dysfunction, including the reduction of LVDP, $+dp/dt_{max}$ and $-dp/dt_{min}$. TAX treatment significantly improved cardiac diastolic dysfunction but did not alter the average HR in isolated heart.

Lactate dehydrogenase is a marker of cellular damage and common disease due to its mass release to plasma during tissue injuries, such as heart failure. CK-MB, which is expressed extensively in cardiac myocyte, was often tested in the serum as an indicator of rhabdomyolysis damage, myocardial damage and acute kidney injury in clinic (*Moghadam-Kia, Oddis & Aggarwal, 2016*). The increase in LDH and CK-MB levels in the organ perfusate after ischemia is a direct evidence of cardiac damage (*Houshmand, Faghihi & Zahediasl, 2009*). In comparison with the control group, LDH and CK-MB activity were significantly increased in the I/R group during myocardial I/R injury. Perfusate LDH and CK-MB activity in the TAX treatment groups, particularly in the TAX 15 μM group, were remarkably reduced compared with those in I/R group, which is consistent with the observation of changing cardiac function parameters. In addition, histopathological examination was implemented to assess the effect of TAX on myocardial morphology. The results of pathomorphological research in the heart samples in the I/R group show acute myocardial damage, and TAX causes favorable morphological changes in the heart during I/R injury. These results supported the potential application of TAX as a cardioprotective agent in myocardial I/R injury.

Under normal conditions, tissues could maintain the balance between generation and clearance of ROS. However, the balance is disrupted during I/R and causes significant increase in ROS (*Becker, 2004*). Excess ROS can oxidize lipids, proteins and DNA, which cause dysfunction of these molecules, resulting in the degeneration of tissue function (*Kleikers et al., 2012*). Minimizing the ROS production is an important strategy to prevent cardiomyocyte I/R injury (*He et al., 2016*). Therefore, the activation of the

anti-oxidant enzyme system is necessary to reduce oxidative stress-induced tissue damage (*Matsushima et al., 2013*). The SOD and GSH-PX rate are used to evaluate tissue per-oxidative injury (*Maciejczyk et al., 2017*). In addition, MDA is an index to evaluate the severity of lipid peroxidation, which is produced by lipid peroxidation, resulting in the destruction of structural proteins and cellular structures (*Pizzimenti et al., 2013*). Our results showed that SOD and GSH-PX activities were conspicuously increased, whereas MDA level was dramatically decreased by TAX, especially in the TAX 15 μM group. Therefore, TAX exhibited the cardioprotective effects by enhancing the antioxidase activity and inhibiting free radical peroxidation.

Mitochondrial damage plays an important role in I/R-induced injury. It is the final arbitrator for I/R-induced cell apoptosis (*Powers et al., 2007*). During ischemia and mainly during the early period of reperfusion, excessive ROS causes myocardial $Ca^{2+}$ overload and the opening of the mitochondrial permeability transition pore, which can reduce mitochondrial function and finally result in an increase in myocardial cell apoptosis (*Garciarena et al., 2011*; *Halestrap & Richardson, 2015*). One of the ways of cell apoptosis is activated by the release of Cyt-c from the mitochondria to the cytosol. In our study, results showed that TAX can weaken the observed increase in the expression of Cyt-c in cytosol. It is very likely that the increased cytosolic content of Cyt-c, which mediates apoptosis, while its expression in mitochondria was not changed (*Lundberg & Szweda, 2004*). Further investigation of the pathological changes in myocardial tissues by TUNEL assays showed different degrees of apoptosis. Results indicated a positive effect of TAX in the inhibition of apoptosis. Therefore, it can make an assumption that down-regulation of Cyt-c result from TAX attenuated apoptotic processes.

As an important mitochondrial regulator during myocardial apoptosis, Bcl-2 exerts anti-apoptotic effects by blocking the release of Cyt-c and reducing caspase activity. Apoptosis-related proteins, caspase 3 and 9, also play crucial roles in apoptosis. The caspase apoptotic pathway responds to death signals by releasing apoptosis-inducing factor from the mitochondria, which were then translocated to the nucleus (*D'Amelio, Sheng & Cecconi, 2012*). In this study, Bcl-2, an anti-apoptotic protein and Bax, a pro-apoptotic protein were used to assess the effects of TAX on cardiomyocytes apoptosis. The result demonstrated that TAX treatment increased the protein expression of Bcl-2, and significantly reduced the Bax expression compared witn the I/R group. Caspase 3 and 9 were tested to measure the apoptotic level in the isolated heart after I/R injury. We found that the increased expression of the active form of caspase 3 and 9 under ischemic conditions and their expression were decreased in the TAX group. Consistent with these results, treatment with TAX significantly decreased myocardial apoptosis by regulating the expression of apoptosis-related proteins, including Bax, Bcl-2, and caspase 3 and 9. These findings suggest that the inhibition of apoptosis is closely related to the underlying beneficial effect of TAX in I/R injury. Cardiomyocytes death occurs during I/R injury by apoptosis, by necrosis and in association with autophagy (*Whelan, Kaplinskiy & Kitsis, 2010*; *Gatica et al., 2015*). In this study, we studied the effect of apoptosis on myocardial injury, but the effect of other pathways on myocardial injury was not excluded, which requires further exploration.

## CONCLUSIONS

In conclusion, TAX exerted cardioprotective effects against I/R injury by inhibiting oxidative stress and cardiac myocyte apoptosis. The underlying mechanism for these phenomena may involve modulation of mitochondrial apoptosis pathway. Our finding provides a novel thought for therapeutic development as an adjuvant therapy to I/R injury.

### Funding

This work was supported by the Outstanding Innovative Talents Support Plan of Heilongjiang University of Chinese Medicine (2012RCD05) and the Outstanding Talent Cultivation Fund Project of Heilongjiang University of Chinese Medicine (2012cj02). The funders had no role in study design, data collection and analysis, decision to publish, or preparation of the manuscript.

### Grant Disclosures

The following grant information was disclosed by the authors:
Outstanding Innovative Talents Support Plan of Heilongjiang University of Chinese Medicine: 2012RCD05.
Outstanding Talent Cultivation Fund Project of Heilongjiang University of Chinese Medicine: 2012cj02.

### Competing Interests

Chunjuan Yang is employed by Harbin Medical University and is a part-time advisor for Beijing Shunyue Technology Co., Ltd. The authors declare they have no competing interests.

### Author Contributions

- Zhenqiu Tang conceived and designed the experiments, performed the experiments, prepared figures and/or tables.
- Chunjuan Yang performed the experiments, authored or reviewed drafts of the paper.
- Baoyan Zuo analyzed the data.
- Yanan Zhang analyzed the data.
- Gaosong Wu contributed reagents/materials/analysis tools.
- Yudi Wang prepared figures and/or tables.
- Zhibin Wang conceived and designed the experiments, approved the final draft.

### Animal Ethics

The following information was supplied relating to ethical approvals (i.e., approving body and any reference numbers):
All animal experiments were approved by College of Pharmacy of Heilongjiang University of Chinese Medicine, Animal Ethics Committee (Approval number: SYXK(hei)-2013-012).

## Data Availability

The raw data is available in the Supplemental Files.

## Supplemental Information

Supplemental information for this article can be found online at http://dx.doi.org/10.7717/peerj.6383#supplemental-information.

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
