# Peer review of "Taxifolin protects rat against myocardial ischemia/reperfusion injury by modulating the mitochondrial apoptosis pathway"

_PeerJ, doi:10.7717/peerj.6383_

## Round 0.1 · original submission · Major Revisions

As you can see, the reviewers made a number of useful suggestions, some of them being very important as they question the validity of your data. So, please, consider them carefully and submit a revised version together with a detailed rebuttal where you explain exactly how (and where in the paper) you have taken them (or not) into account. This rebuttal will be an essential piece of information for us to make a final decision. Please, bear min mind that your revised version will undergo a new round of review by the same or by different reviewers. I cannot, therefore, make any commitment about a final acceptance of your work.

·

Basic reporting

In this study, authors aim to evaluate the protective effects of Taxifolin in myocardial ischemia/reperfusion in rat model. Authors describe that the compound “Tax” modulates
mitochondrial apoptosis pathway to improve the overall cardiac function. This study provides a new insight into the mechanism of this compound on inducing apoptosis.

Experimental design

Well designed, except for the lack of standard Annexin-PI assay for cellular apoptosis.

Validity of the findings

Valid data set

Additional comments

Minor Essential Revisions:
Line 79- recent studies- please provide additional references to support the statement.
Line 117- The hearts was-should be “The hearts were”. (+dp
Line 136- Please define briefly - dp /dtmax and -dp/dtmin), as this is one of the critical parameters of the study.
Line 141-142- “After two hours, the flat tissue blocks were embedded in paraffin µm to make it blocks and then cut into thick tissue sections and stained with H&E”- Poor English need editing.
Line 150- “63 min”, unusual time point, Can authors provide an explanation?
Line 153- what do authors mean by “Lysate? Is it Lysis buffer?”
Line 155- elisa should be “ELISA”.
Line 162- define “specific lysis buffer”
Line 200- heart functional recovery; please spell out the parameters of heart functional recovery here.
Why did the authors ignore the standard qualitative/qualitative assay for cellular Annexin-PI staining of cardiomyocytes in this study?
Figure 4- Arrows indicating the site of tissue damage and edema would add more light to the histological images.

Reviewer 2 ·

Basic reporting

Language of the manuscript lacks clarity, at points it creates ambiguity bordering with nonsense and some phrasing is odd. To give a few examples: Line 46: The sentence “However, blood supply…” lacks a subject and it is not clear what “..will cause additional damage to the heart.” Similarly on line 80: “…TAX involved in the amelioration of cardiac disorders associated with against hypertrophy…” the sentence is incomplete. What is the associated condition and why does it go against hypertrophy? Line 93: “…fed common feedstuff and drank tap water freely …”. What the authors meant is probably feeding the rats with common lab chow and allowing the animals to drink water ad libitum. Certainly using the word “whittled” (line 315) in association with the effect of TAX on apoptosis is very odd. Not only needs the manuscript re-writing, it needs thorough editing by a native speaker. There are too many lines affected by language errors to list.
Setting the language problems aside, the intro provides sufficient background for the study. Although it is not clear, why post-conditioning was used. This is not explained anywhere in the manuscript. Therefore the reader is left wondering why this approach was selected.
Overall structure of the manuscript conforms to standards within the field.
Figures are relevant and of acceptable quality. Labelling is fine, figure legends give redundant information in that they describe what is already apparent from the figure itself.
Raw data are supplied.

Experimental design

The manuscript describes primary original research.
Due to language problems the research question is not explicit besides testing what could be the effect on ischemia/reperfusion damage.
There are several important questions regarding the experimental design and how was the research conducted. Why were the two concentrations of TAX selected? How was the medium containing TAX prepared? Was TAX dissolved in aqueous media directly? If not and stock solution of TAX dissolved in another solvent was used did the controls contain the solvent? Why was post-conditioning used? If selection criteria were applied for the isolated hearts what was the fail rate, i.e. how many hearts failed to meet the criteria, if eight animals were selected per experimental group? At what time point were collected tissue samples that were used for enzyme activity test and immunoblotting? There is no mention of performing subcellular fractionation. Were the proteins, cytochrome c in particular, tested for in total cell lysate? The kits indicated in Materials and Methods generate protein level data in pg/ml, but the corresponding figure gives ng/mg protein. What was the conversion used? The data in Figure 1 show no error bars and do not indicate whether the points shown are mean or median. What was the standard deviation for data shown in Figure 1?
There is no sufficient detail of Methods to allow replication.

Validity of the findings

It is not clear from the manuscript whether the data are robust. That should be made clear in several instances (Methods, Results, Figure legends). The only mention is that there were eight animals per experimental group. Does it mean that each and every heart was subjected to the ischemia protocol, histopathological examination, antioxidant enzymes activity and protein expression? It is possible, however, it is not clearly stated. In any case, the number of samples in each tested group should be given.
Conclusions are based on the data presented and are stated well within the limits of the language used.

Additional comments

Lot of the questions were put forth above. The idea is not irrelevant, however, the manuscript overall is poorly written with insufficient detail given.
For better clarity the main concerns (questions) are repeated here.
1. Why were the two concentrations of TAX selected?
2. How was the medium containing TAX prepared? Was TAX dissolved in aqueous media directly? If not and stock solution of TAX dissolved in another solvent was used did the controls contain the solvent?
3. Why was post-conditioning used?
4. If selection criteria were applied for the isolated hearts what was the fail rate, i.e. how many hearts failed to meet the criteria, if eight animals were selected per experimental group?
5. At what time point were collected tissue samples that were used for enzyme activity test and immunoblotting?
6. There is no mention of performing subcellular fractionation. Were the proteins, cytochrome c in particular, tested for in total cell lysate?
7. The kits indicated in Materials and Methods generate protein level data in pg/ml, but the corresponding figure gives ng/mg protein. What was the conversion used?
8. The data in Figure 1 show no error bars and do not indicate whether the points shown are mean or median. What was the standard deviation for data shown in Figure 1?

Reviewer 3 ·

Basic reporting

In this manuscript, Tang Z et al., aim to demonstrate a function for Taxifolin in protecting against myocardial ischemia/reperfusion injury. Overall, the manuscript is well structured, and provides sufficient evidence to suggest a potential role for Taxifolin in this process. A few minor revisions could help strengthen the quality of the manuscript:

- The legend to figure 1 is missing. A few sentences describing the protocol would be helpful

- The graph in figure 5B is titled MAD, when it’s likely meant to be MDA

- There are several instances of grammatically incorrect sentences. For example, line 57 (‘in the I/R injury”, “cardiomyoctes were observed”), line 81 (“quercetin potentially beneficial”), line 84 (“in present study”), line 226 (“few cells dissolve and degeneration”), line 247 (“may protect mitochondria attenuate apoptosis”), line 252 (“proteins expressions”) , etc. These are not exhaustive, and significantly affect the readability of the manuscript. Perhaps having the text reviewed by a native English speaker would help.

Experimental design

In figure 6, the authors aim to measure the expression of several apoptosis-related proteins. A few concerns regarding this experiment:
- It is unclear from reading the materials section if the commercially available kits used in this experiment are directly measuring protein level (through an ELISA-type method) or if a western-blotting type quantification is used. The authors should clarify the precise method used here
- In panel 6A, the authors claim to be specifically measuring levels of cytosolic-Cytochrome C. However, it is unclear if a specific nuclear-cytosolic fractionation of cell lysate was employed. If so, measuring levels of control proteins that are found only in the cytosol or nucleus will provide confidence in the integrity of the fractionation method

Validity of the findings

Overall, I believe the data presented in this manuscript support the conclusions made by the authors. One point to note however is that I/R-associated injury has been well documented to involve multiple forms of cell death including apoptosis, ferroptosis, and necroptosis. It appears that Taxifolin mediates its protective effect through apoptosis. But the role of other forms of cell death cannot be discounted. The authors should consider adding a brief speculative discussion addressing this point.

---

## Round 0.2 · Major Revisions

As you can see, your revised version was considered still unacceptable by one reviewer. I checked the paper my-self and came to the conclusion that, indeed, your revision was only partial and that additional effort is needed. Please, pay attention to the remarks raised and come back with a revised version where the changes needed have been incorporated in the manuscript (which is what the readers will see). I look forward seeing your new version.

Reviewer 2 ·

Basic reporting

Language is still a major issue. The errors and inconsistencies pointed out in my first review were by no means exhaustive. However, the authors apparently thought so and amended only those. It is not my role to list here every and all errors, therefore just few more examples are given here. Line 53 “…which often cause to death.” Select another verb or re-phrase; line 87 “quercetin has been demonstrated improves post ischemic….” Past perfect or present tense? Cannot be both side by side. Line 94 “..involved in…disorders associated with against cardiac hypertrophy..” Associated with what (an effect? Substance? Protein?) that apparently acts against cardiac hypertrophy. Either complete the sentence or re-phrase it. Line 200 “…homogenized them using …extraction kit…..with a glass homogenizer on ice.” So what was the tool: the kit or the homogenizer? Line 233 “…rehydrated in graded concentration of ethanol.” Did you mean concentration gradient? Or was it a defined concentration of ethanol? Line 265 “At different time points of perfusion, the heart effluents were perfusate.” No clue what information is hidden in this sentence because the heart effluent is in fact the perfusate at all times of the perfusion. What were the authors trying to say here? Line 271 “…CK-MB level was not significantly altered in different group in baseline.” Once again, it is not clear what information is imparted in this sentence.
Let me stress again: this is not an exhaustive list of all unclear statements and it is not my task to find and list all that occur in the manuscript. My persistency in forcing the authors to thoroughly read and correct all unclear and ambiguous language in the manuscript is for the sake of readers of PeerJ.

Experimental design

Methods still lack information to allow replication of the data. Please see also part 4 General comments.

Validity of the findings

no comment

Additional comments

I do understand the answers given by authors to my previous questions and comments. However, to my disappointment only scant information given in the answers actually appears in the manuscript.
Example: Preparation of TAX. Clearly explained but the reader is not told in the manuscript. Even if this communication is posted for readers of PeerJ, I doubt they would look for it. And seek here answers to their questions. So I urge the authors to incorporate these answers into the manuscript. All of them.
I am happy with the explanation of why post-conditioning was employed. It is incorporated in the manuscript.
Selection criteria for animals. Again, please incorporate your answer in the manuscript. The reader should know there were actually 12 animals per group but only eight were used because these met the selection criteria. And of these only some were used for certain post reperfusion testing.
The subcellular fractionation is not sufficiently described in the Methods. I doubt it is one step procedure. What was the g-force used in the centrifugation? Time alone is useless.
Data in figure 2 are now improved. Addition of the error bars raised a question whether there is a significant effect of TAX. The error bars overlap so much that even with reasonable group size (eight) one wonders on the statistics.

Reviewer 3 ·

Basic reporting

It is clear that the authors have taken the reviewer's comments constructively. I have no further suggestions, and believe the manuscript is adequate for publication

Experimental design

No Comment

Validity of the findings

No Comment

Additional comments

No Comment

---

## Round 0.3 · Minor Revisions

I have read your rebuttal and examined your new revised version. I see that most of the changes required have been implemented. However, there remain a few points that require attention. Please, see my notes in the attached PDF, correct what needs to be corrected and submit a new revised version.

---

## Round 0.4 · accepted · Accept

Thank your further correcting your paper.

#